# Unlocking employee creativity: How learning orientation and transformational leadership spark innovation through creative self-efficacy

Chiqing Qian[1], Daisy Mui Hung Kee[1], Biao Zeng[2]*, Hala Najwan Sabeh[3]

1 School of Management, Universiti Sains Malaysia, Penang, Malaysia, 2 Collaborative Innovation Center of Assessment for Basic Education Quality, Beijing Normal University, Beijing, China, 3 Information Technology Department, Tishk International University, Erbil, Iraq

☯ These authors also contributed equally to this work.
* biaozeng@mail.bnu.edu.cn

## Abstract

This study explores the mechanisms through which individual learning orientation (ILO) and transformational leadership (TL) foster employee creativity, with a particular focus on the mediating role of creative self-efficacy (CSE) within small and medium-sized enterprises (SMEs) in China's manufacturing sector. Drawing on a sample of 406 employees, the study employs hierarchical linear modeling (HLM) to analyze the multi-level influences on creativity. The findings indicate that ILO significantly enhances employee creativity, underscoring the importance of continuous personal development and proactive learning in innovation processes. All four dimensions of TL—idealized influence (II), inspirational motivation (IM), intellectual stimulation (IS), and individualized consideration (IC) — are positively associated with employee creativity. CSE is found to mediate the relationship between ILO and creativity, as well as between II, IS, IC, and creativity, although it does not mediate the effect of IM. These results highlight the pivotal role of CSE as a cognitive mechanism through which leadership and learning orientations translate into creative outcomes. The study contributes to the literature on workplace creativity by integrating individual and leadership factors, with psychological resources. Practical implications for SME managers include the cultivation of supportive leadership practices and the promotion of learning-oriented cultures to enhance creative performance. The paper concludes with recommendations for future research directions.

## Introduction

In the context of a global economic downturn, all industries face considerable challenges, and China's manufacturing sector is particularly under tremendous pressure [1]. This stems not merely from the manufacturing SMEs' pivotal role in the Chinese economy, but also from their enduring responsibility for fostering China's

**Data availability statement:** The dataset analysed during the current study is publicly available on the Open Science Framework at the following URL: https://osf.io/5fpbc/.

**Funding:** The author(s) received no specific funding for this work.

**Competing interests:** I have read the journal's policy and the authors of this manuscript have the following competing interests: The authors have declared that no competing interests exist.

transformative innovation and developmental progress [2]. To address the pressing requirements of SMEs, the Chinese government promulgated the "Opinions on Deepening Reform of the Development of the Industrial Workforce" in 2024 [3], which aims to cultivate proficient, innovative industrial workers who significantly contribute to organizational innovation and performance [4]. Obviously, employee creativity — defined as the production of novel and useful products, ideas, or procedures by employees [5] — is not just beneficial; it is essential for survival and sustained success in rapidly evolving industries. Therefore, by examining the antecedents of employee creativity, we aim to provide practical strategies to strengthen employee creativity in SMEs, which in turn promotes innovation and enhances firm performance. Ultimately, this work seeks to foster manufacturing industrial and economic development.

Although previous research has provided many insights into employee creativity [6–9], most of them only conducted explorations of antecedents at a single level, often in isolation. There is a lack of more empirical research results focusing on the influence of antecedents from employees at different levels [10], and few studies have conducted a comprehensive multi-level analysis. Moreover, delving into the antecedents of creativity at the collective level holds even greater significance, particularly in the context of China, which underscores collectivist culture through the cultivation of close and harmonious interpersonal bonds [11,12]. Thus, building upon the insights gleaned from prior research [13–15], this study considers the multifaceted influence of factors at different levels within the organization (i.e., the individual dimension represented by employees, and the team dimension represented by leaders) to bridge existing gaps in academic research and enhance the creativity of employees within SMEs.

Learning and creativity are intrinsically linked. According to Social Cognitive Theory [16], individual learning orientation cultivates adaptability, openness to novel ideas, and perseverance—essential components of creative behavior. While traits like personality (e.g., openness) or intelligence have been associated with creativity, they are generally stable attributes that are less responsive to organizational interventions. In contrast, learning orientation is dynamic and can be nurtured through supportive leadership and organizational culture [17]. Therefore, it is expected to have a positive influence on employee creativity: individuals engage in initial "seed" learning and then receive intuitive signals at work regarding potentially productive elements in their professional areas; these signals lead them to select further elements to learn based on expected value, which may result in creative outcomes [18], even contributing to creative performance at the organizational level. With this premise, we propose individual learning orientation as a prospective direct antecedent to employee creativity.

Transformational leadership is a modern approach that motivates employees beyond their own self-interests for the benefits of their organizations [19]. Such leaders promote risk-taking, challenge conventional thinking, and support employee autonomy, all of which are essential for generating novel ideas and innovative problem-solving [20]. In contrast, other leadership styles like transactional leadership emphasize adherence to performance standards but often lack the psychological

empowerment and visionary behaviors that stimulate creativity [21,22]. Research has also indicated that every dimension of transformational leadership holds significance for fostering employees' creativity [23]. However, intriguingly, most investigations assume an undifferentiated relationship between each dimension and creative outcomes, which means that overwhelming majority operationalize transformational leadership as a molar construct [24], and scant attention has been devoted to specifically or comprehensively examining the influence of distinct dimensions on creative outcomes. In this context, this study will furnish pragmatic insights into the specific dimensions of leadership that require refinement and improvement.

Indeed, based on Social Cognitive Theory of Bandura [16], the relationship between individual learning as personal factors [25], different leadership styles as environmental factors [26] and employee creativity have been widely studied in recent decades. However, research delving into the mechanisms underpinning the correlation between creativity and its antecedents remains underdeveloped [27]. Consequently, inspired by preceding insights from Gong, Huang [26], this study employs creative self-efficacy (CSE) as a mediating construct to probe the antecedent-effect relationships between learning orientation, transformational leadership and creativity from a multi-layered standpoint. Previous studies consistently indicate that CSE serves as a proximal cognitive predictor of creative performance, as individuals with high CSE tend to take creative risks, persevere through challenges, and generate original ideas [28]. As a cognitive mechanism influenced by social and contextual factors, CSE elucidates how leadership, organizational learning environments, or digital transformations manifest in actual creative behaviors [29,30]. In contrast to general motivation or personality traits, CSE is contextually sensitive and actionable, rendering it a precise mediating variable for interventions aimed at enhancing workplace creativity [26], which is also the practical reason for choosing CSE as the mediator.

Overall, our study has four key goals: to scrutinize the impacts of individual learning orientation on employee creativity within a practical context; to delve into the intricate interplay between the four dimensions of transformational leadership and employee creativity in a thorough and comprehensive manner; to elucidate how employee CSE acts as a mediator in the relationship between transformational leadership dimensions, individual learning orientation and employee creativity; and to empirically validate this mediation using a sample drawn from Chinese SMEs, wherein the influence from the collective level cannot be disregarded. SMEs stand as the bedrock of the Chinese economy, wielding a pivotal role in bolstering GDP and propelling China's economic expansion [31]. Consequently, the topic of employee creativity has emerged as a salient and imperative domain of inquiry, given its inherent potential to enhance job performance and directly foster productivity [32,33]. Therefore, our investigation uses multi-dimension, multi-level data to enhance both the conceptual robustness and empirical rigor of the relationship between individual learning orientation, four transformational leadership sub-dimensions, CSE and employee creativity.

## Literature review

### Effects of individual learning orientation on employee creativity

In recent decades, scholars have devoted their scholarly pursuits to the exploration of learning orientation. Despite its extensive examination [34,35], it's noteworthy that learning orientation has been defined differently and has emphasized different facets across diverse studies. It is also worth noting that contemporary scholarly investigations about 'learning orientation' center primarily on empirical inquiries conducted at the organizational level. This implies that numerous scholars have overlooked the fact that organizations inherently lack the innate capability for learning [36]; and learning orientation embedded within an organization is fundamentally rooted in the individual learning endeavors of its constituent members [37,38]. In other words, the true agents of learning and knowledge acquisition within organizations are the individuals themselves. As observed by Kohli, Shervani [39], organizations inherently assimilate knowledge through the collective learning endeavors of their constituents, thus also directly influenced by individual learning. Khedhaouria, Montani [40] further underscored this point by explicitly accentuating learning orientation as the proclivity of employees to prioritize acquiring new skills, mastering unfamiliar scenarios, and honing competencies [41,42]. Therefore, this study adopts the

definition proposed by Gong, Huang [26], wherein learning orientation refers to an individual's concern for and dedication to developing one's competence.

This study employs individual learning orientation to explore the correlation between self-related factors and employee creativity. There are four rationales below underpinning this selection. Foremost and of paramount importance, individual learning orientation, as an innate mindset, serves as a catalyst propelling individuals to augment their proficiency, thus emerging as a pivotal internal impetus for proactive mastery [26]. According to Social Cognitive Theory, individuals can attain knowledge and skills through "enactive mastery experience" [29,43]. Empirical evidence further substantiates that the acquisition of knowledge and skills positively influences creativity [44]. Secondly, individual learning orientation has been linked to intrinsic motivation, a fundamental catalyst for creativity [26]. It functions as a motivational mechanism through which intrinsically motivated employees actively engage in learning endeavors that ultimately yield innovative outcomes [45]. Thirdly, learning orientation and knowledge creation are inexorably intertwined, with the latter being imperative in fueling creativity [46]. While learning orientation encompasses the active pursuit of knowledge, creativity entails the effective application of acquired knowledge [47]. Finally, certain scholars have posited that learning serves as a vital and indispensable wellspring for creativity and innovation [48,49], contributing to a firm's sustenance of competitive advantage over both short and long time horizons [36]. From the foregoing discussions, it can be assumed that individual learning orientation exerts a direct positive influence on employee creativity:

**Hypothesis 1.** Individual learning orientation has a positive direct effect on employee creativity.

### Effects of transformational leadership dimensions on employee creativity

Transformational leadership is widely acknowledged as a pivotal variable within the realm of leadership scholarship, revered for its profound impact on various facets related to organizational performance. Dvir, Eden [50] defined transformational leaders as individuals who wield influence over their subordinates by expanding their horizons and imbuing them with the confidence to transcend predetermined expectations, whether implicit or explicit, set forth in reciprocal agreements. Previous studies have also examined the influence pathway between TL and employee creativity, such as the empirical study of Jyoti and Dev [22]. Henker, Sonnentag [51] has also explored the internal influence mechanism between the two in the form of sequential mediation.

The essence of transformational leadership embodies four cardinal dimensions: idealized behaviors (II), inspirational motivation (IM), intellectual stimulation (IS), and individualized consideration (IC) [19], which have the potential to influence employees' creative comportment through various mechanisms elucidated under social cognitive theory [29,43,52]. Idealized influence, or charisma, reflects a leader's role as a moral and ethical exemplar, fostering trust and admiration that inspire employees to emulate creative behaviors [26]. Inspirational motivation involves articulating a compelling vision that mobilizes employees to exceed expectations, often requiring innovative thinking to meet ambitious goals. Intellectual stimulation encourages critical reflection and challenges existing assumptions, thereby fostering an environment where novel ideas and creative expression can thrive [23]. Individualized consideration emphasizes tailored support and mentorship, enhancing employees' skills, confidence, and creative self-efficacy [53]. Collectively, these dimensions have the conspicuous inter-correlation and function cohesively as an integrated construct of transformational leadership, as evidenced by prior empirical studies [22,54,55].

However, research also has revealed that the four dimensions exhibit overall differentiated content and encompass diverse leader behaviors. For example, the discriminant validity between the transformational leadership dimensions was empirically confirmed in a large sample [56], and some researchers have also found that each dimension has different correlates, for instance, in terms of links with personality traits [57]. Furthermore, several scholars have posited that the precise cognitive and behavioral activities inherent in the leadership process have not received adequate scholarly scrutiny [58,59]. In Li, Zhao [60]'s research, only a pair of transformational leadership dimensions and its correlation with

employee creativity underwent exploration, and they posited that the remaining two dimensions and their relationship with creativity lacked clarity. While the study from Djourova, Rodríguez Molina [24] meticulously gauged all dimensions, its scope was confined to correlational investigations with self-efficacy, neglecting a more in-depth exploration of creative self-efficacy. All of these evidences underscore the importance of individually analyzing each dimension of transformational leadership. Simultaneously, this approach enhances the effectiveness of hypothesis testing regarding the impact of transformational leadership on employee creativity. Hence, our study heeds the call articulated by previous studies to delineate individual dimensions of transformational leadership, as opposed to amalgamating them into a singular overarching scale [61], and posits the following hypothesis:

**Hypothesis 2a.** Idealized influence has a positive direct effect on employee creativity.

**Hypothesis 2b.** Inspirational motivation has a positive direct effect on employee creativity.

**Hypothesis 2c.** Intellectual stimulation has a positive direct effect on employee creativity.

**Hypothesis 2d.** Individualized consideration has a positive direct effect on employee creativity.

## CSE as a mediator

Bandura [43]'s study introduced the notion of self-efficacy within the framework of social cognitive theory (SCT), elucidating its profound impact on the creative process. Bandura proposed that robust self-efficacy is paramount in augmenting creative output, because it influences both the motivation and aptitude of individuals to undertake specific tasks and pursue particular endeavors. Subsequently, Tierney and Farmer [62] embarked on an exploration, defining CSE as the measure of an individual's capacity to generate innovative solutions for an organization. When an individual harbors an innate belief in their capability to manifest exceptional creativity, it denotes a heightened level of CSE [63,64]. As underscored by Lemons [65], creativity springs not merely from competence itself, but from the conviction in one's abilities, thus emphasizing the critical role of self-belief in fostering creative behavior. Consequently, CSE has emerged as a pivotal psychological attribute warranting attention from researchers seeking to unravel the intricacies of enhancing creative performance [66].

Based on the development of SCT, creative self-efficacy beliefs have four primary sources: mastery experiences, vicarious learning, social persuasion and physiological and emotional states [62]. Individual learning orientation appears to positively contribute to the cultivation and perpetuation of CSE through the first, second, and fourth pathways [26]. Multiple plausible justifications exist for this reasoning. Firstly, learning orientation, grounded in an incremental view of aptitude, acknowledges the augmentable nature of skills [67], thereby nurturing the establishment of efficacy beliefs [43]. Secondly, learning orientation facilitates the accrual of triumphant mastery experiences, improving a sense of self-efficacy in fostering innovative outcomes [22,68]. Thirdly, learning orientation proves immensely beneficial for employees when confronting workplace challenges and surmounting the emotional toll of setbacks [69], which aids in bolstering CSE resilience. Lastly, individuals with a learning-centric mindset prioritize skill enhancement over external changes, thereby upholding efficacy beliefs amidst the unpredictable landscape of creativity [26,43]. Payne, Youngcourt [70] posited that a robust learning orientation amplifies specific self-efficacy, while Tierney and Farmer [62] suggested that individuals are more inclined towards creativity when imbued with creative self-efficacy. Then, Gong, Huang [26] raised the possibility that CSE is a mediating variable between learning orientation and employee creativity. In line with that, Qian and Kee [27]'s empirical study clearly confirmed the mediating role of CSE between team learning orientation and creativity. We respond to calls to expand research exploring CSE-related phenomena within conventional or naturalistic work settings [26], and the next hypothesis is:

**Hypothesis 3.** CSE mediates the relationship between individual learning orientation and employee creativity.

Additionally, the concept of self-efficacy, which is lauded as a formidable catalyst for creativity according to social cognitive theory, resonates deeply within the realm of transformational leadership. This resonance stems from the transformative leader's ability to provide employees with steadfast support, genuine encouragement, and indispensable resources [71,72]. CSE blossoms when an individual harbors an internal conviction in their ability to perform with exceptional creativity [30]. Such individuals with elevated CSE are adept at galvanizing motivation, discerning resources, and executing requisite actions to address situational demands. They engage in cognitive problem recognition, brainstorming innovative concepts or solutions, and devote their efforts predominantly to idea generation and prototype production. Consequently, they are well-equipped to navigate specific tasks and surmount challenges encountered during the innovation journey [26,62,73,74]. Moreover, Choi [75]'s study suggested a direct correlation between transformational leadership, CSE, creative performance, and employee creativity. Leaders espousing a transformational leadership style proactively foster creative thinking among their teams, anticipating reciprocal creative engagement. By nurturing their employees' CSE, they effectively cultivate a climate conducive to creativity [50,76].

Despite empirical evidence consistently linking overall transformational leadership to augmented CSE [26,27,36], scant attention has been devoted to discerning the nuances among its dimensions. Van Knippenberg and Sitkin [61] highlighted that the majority of transformational leadership studies neglect to elucidate how each dimension exerts a distinct influence on mediating processes and outcomes. Despite the invaluable insights garnered from prior research, the delineation of boundary conditions in the CSE-creativity nexus remains somewhat restricted [77]. Our endeavor seeks to address this gap in the mechanisms that influence creativity, heeding the recommendations of researchers advocating for the independent exploration of each dimension's effects [24,56,57,61]. Moreover, the potency of these relationships may vary, though we anticipate that all dimensions of transformational leadership will correlate with CSE and employee creativity. Certain dimensions might possess greater potential to influence creativity than others through intermediary mechanisms. Consequently, we expect the four dimensions of transformational leadership as distinct antecedents, rather than a second-order construct, in order to discern which dimensions exert the most profound impact on CSE as a mediator. Hence, we meticulously examine CSE as the mediating mechanism between idealized influence, inspirational motivation, individualized consideration, and intellectual stimulation on followers' creativity:

**Hypothesis 4a.** CSE mediates the effect of idealized influence on employee creativity.

**Hypothesis 4b.** CSE mediates the effect inspirational motivation on employee creativity.

**Hypothesis 4c.** CSE mediates the effect of intellectual stimulation on employee creativity.

**Hypothesis 4d.** CSE mediates the effect of individualized consideration on employee creativity.

The theoretical relationships between the study variables, derived from the literature reviews and assumptions, are illustrated in the conceptual framework (Fig 1).

## Methodology

### Participants and data collection

This study explores the interplay between dimensions of transformational leadership, individual learning orientation, CSE, and employee creativity, focusing on employees within SMEs. The sample is broadly representative of the population of Chinese manufacturing SMEs, with participants drawn from multiple provinces including Jiangsu, Zhejiang, Guangdong, and Shandong — regions known for dense SME manufacturing activity. This geographic diversity helps mitigate regional bias and enhances external validity.

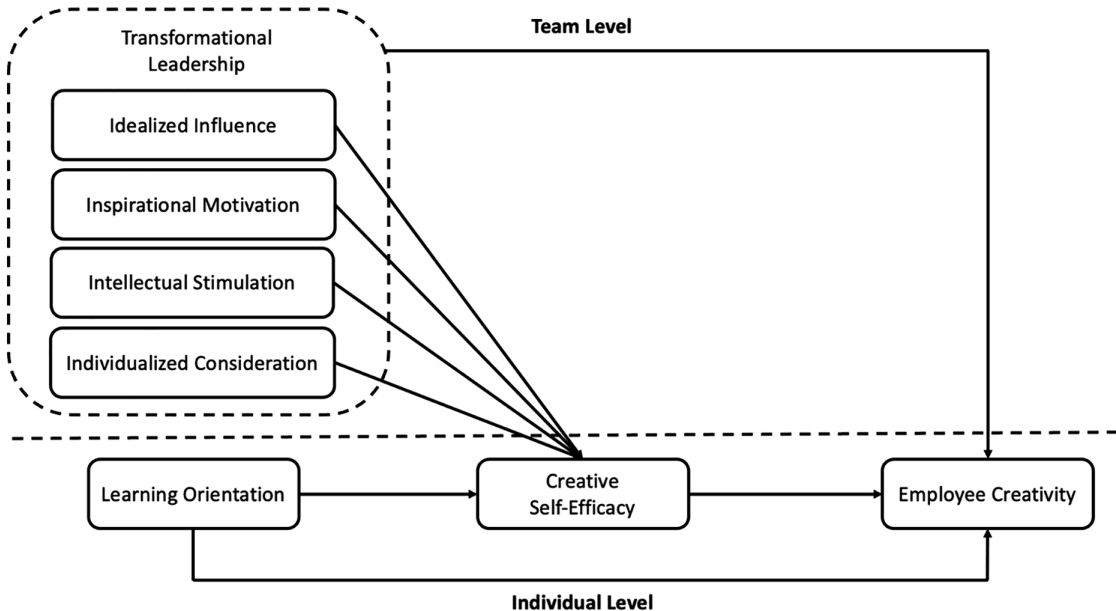

**Fig 1. Theoretical framework.**

Data were gathered from adult employees and supervisors in manufacturing SMEs across China, utilizing a purposive sampling method and a self-administered online questionnaire. The recruitment period lasted from January 1 to March 1, 2024. During this process, the purpose and scope of the study were clearly communicated to ensure that participants were fully informed before taking part. Verbal informed consent was obtained from participants, witnessed and documented by company supervisors. When requested by some participating companies, confidentiality and data use agreements were signed, including a list of employees who had provided informed consent.

Based on the list of participants, the questionnaire invitation and link were privately distributed via WeChat to employees and their direct supervisors. All survey instruments were administered in Chinese, the respondents' native language. To promote transparency and participation, a cover letter was included on the first page of the online questionnaire, stating the purpose of the study, providing guidance, assuring confidentiality, and listing the researcher's contact information.

After two weeks, the researcher reviewed and recorded the responses received. Follow-up communication was conducted in late January to capture any additional submissions. During preliminary data screening, extreme values and invalid responses—such as those completed in an unusually short time—were excluded. Of the 500 questionnaires distributed, 406 were effectively completed and returned, yielding a response rate of 81.2%. Table 1 presents the demographic profile of the respondents.

## Measurement of variables

**Main variables.** This research evaluated main variables using a mixture of seven-point Likert scale and five-point scale. The instruments used in this research were involved in and adopted from previous studies. We used back-to-back translation method to ensure the accuracy of language between English and Chinese.

Firstly, to measure the employees' subjective experiences of creativity, 3 items from Oldham and Cummings [5]'s scale were adopted. Hence, employees' creativity from Chinese SMEs is measured with a 5-point Likert scale of frequency, ranging from 1 (never) to 5 (always). The Cronbach's alpha coefficient for this measure was 0.91.

**Table 1. Profiles of respondents.**

| Demographic Variable | Category | Frequency | Percentage |
|---|---|---|---|
| **Age** | 20 and below | 4 | 1.0 |
| | 21–30 | 178 | 43.8 |
| | 31–40 | 68 | 16.7 |
| | 41–50 | 87 | 21.4 |
| | 51–60 | 67 | 16.5 |
| | 61 and above | 2 | 0.5 |
| **Gender** | Male | 153 | 37.7 |
| | Female | 253 | 62.3 |
| **Education** | Primary school | 11 | 2.7 |
| | Middle school | 42 | 10.3 |
| | High school | 53 | 13.1 |
| | Bachelor's degree | 222 | 54.7 |
| | Master/Ph.D. degree | 78 | 19.2 |
| **Job Tenure** | Less than 1 year | 10 | 2.5 |
| | 1 to 5 years | 6 | 1.5 |
| | 6 to 10 years | 13 | 3.2 |
| | More than 10 years | 377 | 92.9 |
| **Team Size** | 5 people | 180 | 44.3 |
| | 6 people | 126 | 31.0 |
| | 7 people | 49 | 12.1 |
| | 8 people | 32 | 7.9 |
| | 9 people | 9 | 2.2 |
| | 10 people | 10 | 2.5 |
| | Above 10 people | 0 | 0 |

n = 406.

For evaluating individual learning orientation in this study, we employed six items adapted from Gong, Huang [26]. Responses were collected using a seven-point Likert scale, ranging from 1 (strongly disagree) to 7 (strongly agree), and the scores were averaged to create a composite index. The Cronbach's alpha for this measure was 0.94.

To assess four dimensions from the construct of transformational leadership, the study utilized the Multifactor Leadership Questionnaire (MLQ) Form 5X-Short [78]. This instrument comprises 20 items distributed across four subscales: 'Idealized Influence', 'Inspirational Motivation', 'Intellectual Stimulation', and 'Individualized Consideration'. Each item was rated on a seven-point Likert scale, from 1 (strongly disagree) to 7 (strongly agree). The Cronbach's alpha coefficients for the subscales were 0.89, 0.91, 0.90, and 0.93, respectively.

Finally, as the mediating variable, CSE was assessed using a three-item scale developed by Tierney and Farmer [62]. Responses were recorded on a five-point Likert scale, ranging from 1 (not at all) to 5 (to a great extent). The Cronbach's alpha for employee CSE in the current study was 0.93 [62].

**Control variables.** In testing the hypotheses, we accounted for variables such as age, gender, education level, job tenure, and team size. Education level and job tenure were controlled for, as they may impact the domain-specific knowledge and expertise crucial for fostering creativity [26]. Previous research also indicates that team size can affect group dynamics [79] and has been linked to employee creativity in prior studies [25].

## Data analysis techniques

The data were mainly analyzed by multilevel analysis using HLM 6.08 software. After completing the data screening and deriving descriptive statistics, the study used hierarchical linear modeling (HLM) to examine the direct relationship between the proposed variables. HLM serves as an analytical framework specifically designed for datasets featuring nested sources of variability, where micro units are embedded within macro units [80]. The data in this study were standardized and organized into two levels: Level 1 variables related to individual characteristics and Level 2 variables pertaining to the 70 groups [81]. The analysis was conducted in a two-step process: first, the outer measurement model was evaluated to determine its validity; second, the hierarchical linear model was assessed to test the proposed hypotheses. More importantly, this study used Monte Carlo simulation to test the mediation effect [82].

## Results

### Assessment of the measurement model

Table 2 displays the means, standard deviations, reliability and correlations of the variables included in our study. Individual learning orientation was positively related to employee creativity ($r = 0.532$, $p < 0.01$). All the transformational leadership dimensions, including idealized influence (II), inspirational motivation (IM), intellectual stimulation (IS) and individualized consideration (IC), were positively related to employee creativity ($r = 0.510$, $p < 0.01$; $r = 0.534$, $p < 0.01$; $r = 0.498$, $p < 0.01$, and $r = 0.471$, $p < 0.01$, respectively). CSE was also positively related to employee creativity ($r = 0.639$, $p < 0.01$).

### Assessment of the hierarchical linear model

Prior to conducting cross-level analysis, it is imperative to assess the suitability of aggregating the Transformational Leadership (TL) dimensions to the organizational level, considering the data was collected from individual employees. To evaluate the appropriateness of variable aggregation, this study employed the $r_{wg}$ statistic, as recommended by James, Demaree [83]. The computed $r_{wg}$ values for group cohesion in the TL dimensions were 0.749, 0.914, 0.937, and 0.970, respectively, all surpassing the 0.70 threshold [84]. These results indicate a high level of consensus among raters within each group, thereby supporting the aggregation of TL dimensions to the organizational level for subsequent cross-level analysis.

**Results of hierarchical linear modeling for direct effects.** In accordance with the guidelines set forth by Raudenbush and Bryk [85], a comprehensive multi-level analysis should include four distinct sub-models: the Null Model (Model I), Random Coefficient Model (Model II), Intercepts as Outcomes Model (Model III), and Slopes as Outcomes Model (Model IV). Given that our study did not investigate the moderating effects of team-level contextual variables, we

**Table 2. Means, standard deviations, reliability, and correlations between main variables.**

| Variable | M | SD | α | 1 | 2 | 3 | 4 | 5 | 6 |
|---|---|---|---|---|---|---|---|---|---|
| **Transformational Leadership** | | | | | | | | | |
| Idealized Influence (II) | 4.67 | 1.65 | 0.893 | 1 | | | | | |
| Inspirational Motivation (IM) | 4.74 | 1.52 | 0.910 | 0.880** | 1 | | | | |
| Intellectual Stimulation (IS) | 4.66 | 1.65 | 0.901 | 0.863** | 0.895** | 1 | | | |
| Individualized Consideration (IC) | 4.53 | 1.66 | 0.932 | 0.845** | 0.858** | 0.883** | 1 | | |
| **Individual Learning Orientation** | 5.23 | 1.39 | 0.938 | 0.646** | 0.633** | 0.618** | 0.593** | 1 | |
| **Creative Self-Efficacy** | 3.51 | 1.05 | 0.933 | 0.573** | 0.595** | 0.589** | 0.566** | 0.748** | 1 |
| **Employee Creativity** | 3.35 | 1.13 | 0.911 | 0.510** | 0.534** | 0.498** | 0.471** | 0.532** | 0.639** |

** Correlation is significant at the 0.01 level, * Correlation is significant at the 0.05 level.

excluded the Slopes as Outcomes Model from our analysis. Consequently, our multi-level analysis was limited to the first three models.

Model I, namely Null Model, which was used to determine the proportion of variance in employee creativity attributable to group-level differences and to evaluate the intraclass correlation coefficient (ICC, $\rho$) of the dependent variable, thereby assessing the appropriateness of the between-group variance component for multi-level analysis. As detailed in Table 3, the within-group variance ($\sigma^2$) and between-group variance ($\tau_{00}$) for employee creativity were calculated at 0.546 and 0.763, respectively ($\chi2 = 619.461$, df = 69, p < 0.001). The between-group variance component is significantly distinct from zero, indicating substantial variability in employee creativity levels across different supervisors. Additionally, the ICC values calculated by $\sigma^2$ and $\tau_{00}$ for employee creativity were substantial, reflecting that 58.289% of the total variance in creativity can be attributed to the upper-level factor, namely the team level led by direct managers. An ICC greater than 0.138 suggests a high degree of intraclass correlation, while an ICC exceeding 0.059 indicates the need for multi-level analysis [86]. These all justified the use of a multi-level model in this study to avoid biased interpretations of the data.

Then, the Random Coefficient Regression Model (Model II) was employed to investigate the direct effects of individual-level variables on creativity, focusing solely on the first-level independent variables, while the second level was represented by a zero model. In this framework, the regression coefficient at the first level includes both the intercept and the slope term from the second-level regression model, with both components treated as random effects. The primary aim of this analysis was to assess the significance of the intercept and slope within the first-level regression model, particularly concerning the independent variable of individual learning orientation (ILO) in this study. The findings provided support for Hypothesis 1, revealing a significant relationship between individual learning orientation and employee creativity ($\gamma = 0.265$, s.e. = 0.062, t = 4.269, df = 69, p < 0.001). These results align with the Hierarchical Linear Modeling (HLM) analysis presented in Table 3.

To further elucidate the primary effect of the second-level variable, specifically the dimensions of transformational leadership, on employee creativity, an Intercepts as Outcomes Model (Model III) was employed based on preceding analyses. Transformational leadership is delineated into four sub-dimensions: idealized influence (II), inspirational motivation (IM), intellectual stimulation (IS), and individualized consideration (IC). The outcomes of the Intercepts as Outcomes Model analysis are also detailed in Table 4. To evaluate Hypothesis 2a, 2b, 2c and 2d, we performed separate regressions

**Table 3. HLM results of null model and random coefficient model.**

| Variable | Employee Creativity | | | | | |
|---|---|---|---|---|---|---|
| | Model I | | | Model II | | |
| **Fixed Effect** | $\gamma$ | s.e. | t | $\gamma$ | s.e. | t |
| $\gamma_{00}$ | 3.349 | 0.110 | 30.451*** | 3.349 | 0.110 | 30.445*** |
| | Individual-level | | | | | |
| Age | | | | 0.005 | 0.039 | 0.123 |
| Gender | | | | −0.152 | 0.087 | −1.741 |
| Education | | | | 0.153 | 0.096 | 1.591 |
| Job Tenure | | | | −0.117 | 0.080 | −1.460 |
| Individual Learning Orientation | | | | 0.265 | 0.062 | 4.269*** |
| **Random Effect** | v.c. | $\chi^2$ | p | v.c. | $\chi^2$ | p |
| $\tau_{00}$ | 0.763 | 619.461 | < 0.001 | 0.771 | 672.320 | < 0.001 |
| $\sigma^2$ | 0.546 | | | 0.503 | | |
| **Deviance** | 1060.011 | | | 1047.636 | | |

$\gamma$ = Parameter Estimate; s.e. = Standard Error; v.c. = Variance Component; t = T-Ratio; *p < 0.05, **p < 0.01, ***p < 0.001.

**Table 4. HLM results of intercepts as outcomes model.**

| Variable | Employee Creativity | | | | | | | | | | | |
|---|---|---|---|---|---|---|---|---|---|---|---|---|
| | Model III | | | | | | | | | | | |
| **Fixed Effect** | γ | s.e. | t | γ | s.e. | t | γ | s.e. | t | γ | s.e. | t |
| $\gamma_{00}$ | 3.350 | 0.087 | 38.572*** | 3.350 | 0.086 | 38.821*** | 3.349 | 0.087 | 38.402*** | 3.350 | 0.087 | 38.584*** |
| | Individual-level | | | | | | | | | | | |
| Age | 0.030 | 0.040 | 0.743 | 0.030 | 0.040 | 0.743 | 0.030 | 0.040 | 0.743 | 0.030 | 0.040 | 0.743 |
| Gender | −0.157 | 0.098 | −1.600 | −0.157 | 0.098 | −1.600 | −0.157 | 0.098 | −1.600 | −0.157 | 0.098 | −1.600 |
| Education | 0.141 | 0.094 | 1.502 | 0.141 | 0.094 | 1.502 | 0.141 | 0.094 | 1.502 | 0.141 | 0.094 | 1.502 |
| Job Tenure | −0.131 | 0.097 | −1.356 | −0.131 | 0.097 | −1.356 | −0.131 | 0.097 | −1.356 | −0.131 | 0.097 | −1.356 |
| | Team-level | | | | | | | | | | | |
| Team Size | 0.065 | 0.075 | 0.863 | 0.066 | 0.071 | 0.936 | 0.021 | 0.078 | 0.272 | 0.023 | 0.074 | 0.308 |
| Idealized Influence (II) | 0.348 | 0.060 | 5.802*** | | | | | | | | | |
| Inspirational Motivation (IM) | | | | 0.381 | 0.059 | 6.478*** | | | | | | |
| Intellectual Stimulation (IS) | | | | | | | 0.345 | 0.059 | 5.862*** | | | |
| Individualized Consideration (IC) | | | | | | | | | | 0.344 | 0.054 | 6.342*** |
| **Random Effect** | v.c. | $\chi^2$ | p | v.c. | $\chi^2$ | p | v.c. | $\chi^2$ | p | v.c. | $\chi^2$ | p |
| $\tau_{00}$ | 0.451 | 383.979 | < 0.001 | 0.444 | 377.450 | < 0.001 | 0.460 | 396.730 | < 0.001 | 0.454 | 390.601 | < 0.001 |
| $\sigma^2$ | 0.538 | | | 0.538 | | | 0.537 | | | 0.537 | | |
| **Deviance** | 1038.729 | | | 1037.676 | | | 1039.513 | | | 1038.824 | | |

γ = Parameter Estimate; s.e. = Standard Error; v.c. = Variance Component; t = T-Ratio; *p < 0.05, **p < 0.01, ***p < 0.001.

of employee-rated transformational leadership dimensions on employee creativity, incorporating control variables. The analysis supported Hypothesis 2, demonstrating that all four dimensions of transformational leadership positively correlate with employee creativity: idealized influence (γ = 0.348, s.e. = 0.060, t = 5.802, df = 67, p < 0.001), inspirational motivation (γ = 0.381, s.e. = 0.059, t = 6.478, df = 67, p < 0.001), intellectual stimulation (γ = 0.345, s.e. = 0.059, t = 5.862, df = 67, p < 0.001), and individualized consideration (γ = 0.344, s.e. = 0.054, t = 6.342, df = 67, p < 0.001).

**Results of hierarchical linear modeling for mediating effects.** As previously noted, the indirect effects were evaluated using the Monte Carlo method to construct confidence intervals, thereby determining whether the mediated effects are significantly distinct from zero [82]. Even though previous studies have shown that transformational leadership affects employee creativity through CSE [27], whether the four sub-dimensions of transformational leadership (II, IM, IS and IC) have effects separately requires the use of more precise Monte Carlo estimation to test the mediation path. Results for all mediation effects can be seen in Table 5, where the significant indirect effects are marked in boldface.

First, the results of this study support the indirect effect of CSE as a mediating variable between employee learning orientation and creativity (Hypothesis 3). As shown in Table 5, CSE does mediate the relationship between personal learning orientation and creativity (lower limit [LL] = 0.012; upper limit [UL] = 0.068)

Additionally, we tested whether CSE mediated between four TL dimensions and employee creativity. Their role as partial mediators was supported for three TL dimensions (II, IS and IC). The effect of idealized influence (II) on employee creativity via CSE was significant (Hypothesis 4a, LL = 0.009; UL = 0.118), as was the indirect effect of intellectual stimulation (IS) on employee creativity via CSE (Hypothesis 4c, LL = 0.011; UL = 0.138). The same applies for the indirect effects of individualized consideration (IC) on creativity (Hypothesis 4d, LL = 0.005; UL = 0.046). However, Hypothesis 4b is

**Table 5. Estimates and Monte Carlo confidence intervals for the indirect effects.**

| Mediation Paths | Estimate | SE | Bias-Corrected 95% CI | |
|---|---|---|---|---|
| | | | Lower | Upper |
| **Individual Learning Orientation–CSE–Employee Creativity** | **0.036** | **0.017** | **0.012** | **0.068** |
| **Idealized Influence (II)–CSE–Employee Creativity** | **0.040** | **0.034** | **0.009** | **0.118** |
| Inspirational Motivation (IM)–CSE–Employee Creativity | 0.027 | 0.056 | −0.028 | 0.129 |
| **Intellectual Stimulation (IS)–CSE–Employee Creativity** | **0.048** | **0.043** | **0.011** | **0.138** |
| **Individualized Consideration (IC)–CSE–Employee Creativity** | **0.020** | **0.012** | **0.005** | **0.046** |

5,000 bootstrap samples; CI = confidence interval; SE = standard error. Significant indirect effects values are in boldface, where the CI does not include zero.

rejected because the indirect effect was not significant — the confidence interval contains zero, which means CSE does not mediate the relationship between inspirational motivation (IM) and employee creativity.

## Discussion

This study accomplished four research objectives: empirically investigated the correlation between individual learning orientation and employee creativity; delved into the impact of the four sub-dimensions of transformational leadership (II, IM, IS and IC) on employee creativity with a more refined hierarchical structure compared to previous research; evaluated CSE as a mediator in the relationship between individual learning orientation, transformational leadership dimensions and employee creativity; and examined the drivers behind employee creativity using a sample from Chinese SMEs. Subsequently, we will discuss these findings in the context of existing research results.

### Theoretical implications

Firstly, this study contributes to the employee creativity literature by examining the boundary conditions of learning orientation on individual creativity among Chinese SMEs. Our results about the significant positive correlation between individual learning orientation and employee creativity unravel the intrinsic element that improves employee creativity. Interpreted from the perspective of incremental creativity [87], employee learning orientation is more likely to enhance employee creativity over time, because an individual's actual learning orientation has a stronger effect on creativity than does an artificially derived (manipulated) learning orientation [26]. In addition, this study emphasizes that learning orientation in organizations fundamentally originates from individual employees' learning [37], and follows Gong, Huang [26]'s tendency in the investigation to focus on individual-level learning orientation when exploring employee innovation. Therefore, this research also addresses a gap in the literature by focusing on individual-level learning orientation, moving beyond the more commonly studied topic of organizational learning.

Moreover, in recent years, several quantitative research have investigated transformational leadership within Chinese enterprises. However, the scope of these findings remains somewhat limited in terms of generalizability. Wu [88] underscored the research gap in scholarly inquiries pertaining to leadership within SMEs relative to their counterparts in the realm of high-tech and internationalized enterprises. The present study helps fill this gap in knowledge, showcasing a favorable association between transformational leadership and creativity among employees in SMEs. It is worth noting that the second conclusion (Hypothesis 2a, 2b, 2c and 2d) complements the finding of Li, Zhao [60], who only partially studied and analyzed the impact of transformational leadership dimensions in the field of creativity's antecedents. Meanwhile, this study not only measured the impact of all TL sub-dimensions on employee creativity, but also further developed Djourova, Rodríguez Molina [24]'s finding by conducting more deeply exploring the role of all the dimensions in the mediating mechanism. As a result, this study aligns with previous findings that illustrate a positive correlation between

dimensions of transformational leadership and creativity [26,89–91], further extrapolating this association to the specific landscape of Chinese enterprises. In the context of China, characterized by its collectivist ethos, transformational leadership presents different effects and practical values distinct from its Western counterparts [92]. Hence, leaders esteemed for their exemplary principles and unwavering ethical integrity are predisposed to nurture employee creativity by instilling a profound sense of pride, motivation, and confidence, which can be achieved through deliberate acts of intellectual stimulation and tailored support [91,93].

Finally, this study extends the research on employee creativity by introducing a valuable intermediary, namely CSE, to elucidate the distinct pathways through which learning orientation and transformational leadership independently bolster employee creativity. Pragmatically, our investigation fortifies the essential role of personal learning in bridging the gap in individual creativity through the cultivation and refinement of CSE, which is an implication in harmony with prior scholarship [26]. Moreover, this paper focuses on the more comprehensive and detailed mediating role played by CSE in the mechanism of transformational leadership's impact on employee creativity, of which results indicates that CSE can significantly and positively mediate the effects of three transformational leadership's sub-dimensions (II, IS and IC) on individual creativity, except inspirational motivation (IM). This may be because the inspirational incentives issued by transformational leaders have a very strong direct impact on employee intentions and behaviors, and the indirect effect of CSE as a mediating variable on employee creativity may be minimal, making the mediating effect insignificant. This is also confirmed by the fact that among all TL dimensions, IM has the strongest correlation with CSE and EC in direct effects. Hence, our research endeavors to respond to the scholarly calls for delving into the mediating factors influencing the correlation between creativity and its antecedents within the sphere of SMEs [94–96]. The outcomes of this study offer valuable insights to SME managers, enabling them to refine their management strategies by harnessing CSE as a catalyst for nurturing employee creativity and innovation.

In burgeoning economies such as China, organizations are committed to fostering employee creativity and seek leaders endowed with visionary acumen. Our investigation delves into the potential of individual learning orientation and transformational leadership dimensions to cultivate CSE, thereby fostering employee creativity that is an imperative for attaining sustainable competitive advantage in the contemporary business landscape.

## Practical implications

This study adopts a cross-level methodology to study employee creativity, culminating in theoretical deductions, and now endeavors to develop managerial recommendations rooted in these theoretical frameworks.

In today's fiercely competitive landscape of SMEs, innovative employees emerge as linchpins for sustainable competitive advantage [64,97,98]. The study reveals that individuals inclined towards learning orientation possess the capacity to amplify their creative faculties. As widely acknowledged, the learning behavior fundamentally resides within each individual entity. Within Chinese manufacturing SMEs, given the profound correlation between individual learning orientation and employee creativity, it behooves employees to actively cultivate an awareness of lifelong learning. And then individuals can enhance their own capabilities in the workplace, including personal creativity, by augmenting their competencies, accruing experiential wisdom, and pursuing further education. Similarly, for leaders within SMEs, this study advocates for a conscientious consideration of candidates' predisposition towards learning orientation during the recruitment process. Such hiring and management discernment can yield dividends over time by fostering a heightened level of creativity among employees and bolstering their creative performance. Furthermore, against the backdrop of collectivism entrenched within the Chinese context, the establishment of environments conducive to learning assumes paramount importance. By fostering an ecosystem that nurtures proactive knowledge acquisition and dissemination among employees, organizations can be more likely to succeed [99–101]. Through such endeavors, employee creativity stands poised to flourish, underpinned by the acquisition of enriched knowledge and profound insights, thereby propelling organizational performance to greater heights [102].

In addition, it is recommended that corporate managers must grasp the correlation between leadership style and the cultivation of employee creativity, which helps supervisors to create a work environment that fosters employee creativity, as highlighted in Henker, Sonnentag [51]'s study. The research findings underscore the paramount importance of adopting transformational leadership practices within Chinese SMEs, owing to its congruence with the developmental needs of such enterprises. This leadership paradigm exhibits multifaceted advantages that are exceptionally conducive to fostering creative behaviors and nurturing employees' self-efficacy perceptions. Furthermore, the outcomes are in alignment with prior research, affirming that each sub-dimension of transformational leadership significantly contributes to the actualization of employees' creativity within SMEs, given their motivational and cognitive capabilities deemed indispensable for fostering and evaluating creative performance [103]. Drawing from the four facets of transformational leadership—idealized behaviors (II), inspirational motivation (IM), intellectual stimulation (IS), and individualized consideration (IC) — this study propounds several recommendations to enhance the implementation of this leadership approach, thereby empowering SMEs to effectively realize their objectives of management and development. These recommendations may be encapsulated as follows: (a) Leaders must exemplify personal attributes such as focus, adaptability, and risk-taking, ensuring alignment between rhetoric and action. Additionally, they should lead by example in managing workloads [104], while fostering empathetic communication and fostering a culture of collaboration and solidarity within the team to uplift team morale; (b) Leaders must articulate a clear vision and mission, underscored by consistent values, exacting standards, and a profound sense of purpose, coupled with unwavering confidence in both themselves and their subordinates, thereby earning the trust and esteem of their followers [27]; (c) Leaders must actively pursue novel ideas and methodologies to enhance task efficacy, galvanizing team motivation and enthusiasm through charismatic influence, exemplary conduct, inspiration, and the provision of creative encouragement and intellectual stimulation; (d) Leaders must acknowledge the intrinsic value of human capital in fostering advancement and progress, attending to individual needs, harnessing their potential, and facilitating their personal growth and achievement [105].

Finally, the explanation for the positive mediating function of CSE on employee creativity could be ascribed to one's intrinsic yearning for creativity, which is consistent with one of the findings in this study. CSE has emerged as a pivotal self-regulatory mechanism for motivating and sustaining efforts in fostering creativity [25]. Particularly under the negative impact of the economic downturn, where organizational operations have not yet fully recovered to their previous normal levels (e.g., teamwork), and where the physical and mental well-being of workers is impacted by stressors (e.g., unemployment stress) [106,107], it becomes imperative for managers within the manufacturing industry to deepen their comprehension of the potential interconnections among learning orientation, transformational leadership, CSE, and employee creativity. Hence, we advocate that SMEs redouble their endeavors in nurturing their followers and imbuing them with confidence to augment their creative prowess through the adoption of transformational leadership practices. Furthermore, enterprises can facilitate their employees in embracing a consciousness of continuous learning and personal growth, thereby adeptly undertaking innovative tasks and honing the requisite creative aptitudes, by bolstering their confidence and involvement in creative pursuits.

The significance of SMEs in China's economic trajectory cannot be overstated. Hence, it behooves company leaders and scholars to discern the latent potential within SMEs in China to yield innovative outcomes and to foster an ecosystem conducive to creativity within such enterprises.

## Limitations and future research

This study has several limitations. Firstly, the survey relied upon participants' self-reported data. Despite efforts to mitigate and control for inherent biases, this approach increases the risk of common method variance, which cannot be entirely eliminated. Secondly, given the study's applicability solely to the Chinese manufacturing realm, researchers are urged to extend their data collection efforts to encompass diverse industrial sectors within China, thus enabling the broader generalization of their findings beyond manufacturing SMEs. Lastly, the study's adoption of a cross-sectional research

framework restricted the measurement of variables to a specific point in time, impeding the unequivocal determination of causal relationships among the variables scrutinized. Future inquiries should embrace experimental and longitudinal research methodologies to elucidate the causal nexus between learning orientation, transformational leadership, CSE, and employee creativity, thereby engendering more reliable conclusions.

## Conclusions

In summary, this study endeavors to elucidate the impact of learning orientation and transformational leadership sub-dimensions on employee creativity within Chinese SMEs, with CSE serving as the mediator and a particular emphasis on individual-contextual dynamics. This scholarly exploration aims to enrich the existing discourse on employee creativity. Despite the considerable investments that organizations have devoted to honing employee capabilities, clear practical guidance still needs to be explored on the intricate interplay of antecedent variables spanning diverse strata and the complex mechanisms through which antecedent variables at different levels affect employee creativity. Consequently, the insights gleaned from this academic research are poised to stimulate the intellectual pursuit of corporate researchers, impelling them to delve deeper and unveil fresh perspectives of pragmatic significance, particularly for SMEs.

## Author contributions

**Conceptualization:** Chiqing Qian, Daisy Mui Hung Kee, Biao Zeng, Hala Najwan Sabeh.

**Data curation:** Chiqing Qian.

**Formal analysis:** Chiqing Qian.

**Investigation:** Chiqing Qian.

**Methodology:** Chiqing Qian.

**Project administration:** Chiqing Qian, Daisy Mui Hung Kee, Biao Zeng, Hala Najwan Sabeh.

**Resources:** Chiqing Qian.

**Software:** Chiqing Qian.

**Supervision:** Daisy Mui Hung Kee, Biao Zeng.

**Validation:** Chiqing Qian.

**Visualization:** Chiqing Qian, Daisy Mui Hung Kee, Biao Zeng, Hala Najwan Sabeh.

**Writing – original draft:** Chiqing Qian.

**Writing – review & editing:** Chiqing Qian, Daisy Mui Hung Kee, Biao Zeng, Hala Najwan Sabeh.

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
