## [Decision Letter · Decision Letter 0]

17 Jul 2025

Dear Dr. Zeng,

Thank you for submitting your manuscript to PLOS ONE. After careful consideration, we feel that it has merit but does not fully meet PLOS ONE’s publication criteria as it currently stands. Therefore, we invite you to submit a revised version of the manuscript that addresses the points raised during the review process.

We look forward to receiving your revised manuscript.

Kind regards,

Cheong Kim

Academic Editor

PLOS ONE

**Journal Requirements:**

1. When submitting your revision, we need you to address these additional requirements. Please ensure that your manuscript meets PLOS ONE's style requirements, including those for file naming. The PLOS ONE style templates can be found at https://journals.plos.org/plosone/s/file?id=wjVg/PLOSOne_formatting_sample_main_body.pdf and https://journals.plos.org/plosone/s/file?id=ba62/PLOSOne_formatting_sample_title_authors_affiliations.pdf 2. You indicated that ethical approval was not necessary for your study. We understand that the framework for ethical oversight requirements for studies of this type may differ depending on the setting and we would appreciate some further clarification regarding your research. Could you please provide further details on why your study is exempt from the need for approval and confirmation from your institutional review board or research ethics committee (e.g., in the form of a letter or email correspondence) that ethics review was not necessary for this study? Please include a copy of the correspondence as an "Other" file. 3. We note that your Data Availability Statement is currently as follows: All relevant data are within the manuscript and its Supporting Information files. Please confirm at this time whether or not your submission contains all raw data required to replicate the results of your study. Authors must share the “minimal data set” for their submission. PLOS defines the minimal data set to consist of the data required to replicate all study findings reported in the article, as well as related metadata and methods (https://journals.plos.org/plosone/s/data-availability#loc-minimal-data-set-definition). For example, authors should submit the following data: - The values behind the means, standard deviations and other measures reported;- The values used to build graphs;- The points extracted from images for analysis. Authors do not need to submit their entire data set if only a portion of the data was used in the reported study. If your submission does not contain these data, please either upload them as Supporting Information files or deposit them to a stable, public repository and provide us with the relevant URLs, DOIs, or accession numbers. For a list of recommended repositories, please see https://journals.plos.org/plosone/s/recommended-repositories. If there are ethical or legal restrictions on sharing a de-identified data set, please explain them in detail (e.g., data contain potentially sensitive information, data are owned by a third-party organization, etc.) and who has imposed them (e.g., an ethics committee). Please also provide contact information for a data access committee, ethics committee, or other institutional body to which data requests may be sent. If data are owned by a third party, please indicate how others may request data access. 4. If the reviewer comments include a recommendation to cite specific previously published works, please review and evaluate these publications to determine whether they are relevant and should be cited. There is no requirement to cite these works unless the editor has indicated otherwise. 

Reviewers' comments:

Reviewer's Responses to Questions

**Comments to the Author**

1. Is the manuscript technically sound, and do the data support the conclusions?

Reviewer #1: Yes

Reviewer #2: Yes

Reviewer #3: Yes

2. Has the statistical analysis been performed appropriately and rigorously?

Reviewer #1: Yes

Reviewer #2: Yes

Reviewer #3: I Don't Know

3. Have the authors made all data underlying the findings in their manuscript fully available?

Reviewer #1: Yes

Reviewer #2: Yes

Reviewer #3: Yes

4. Is the manuscript presented in an intelligible fashion and written in standard English?

Reviewer #1: Yes

Reviewer #2: Yes

Reviewer #3: Yes

**Reviewer #1: ** The paper touches on an interesting topic and is supported by a valuable and well-developed empirical study. However, it takes a very narrow view of the problem studied. It basically shows several variables and justifies why it makes sense to study the relationship between these variables. However, it does not describe why it is important to study the employee creativity. What does this study contribute to science and society? Why is the employee creativity related to the individual learning orientation and not to other variables? What does this contribute? And the same when talking about employee creativity and transformational leadership. The abstract and the introduction would have to be completely redone.

Once we get into the variables chosen and their relationship with employee creativity (literature review section), we can see that the relationships between variables are justified by citing some references in the literature, but the literature review does not go very far. For example, when studying the relationship between transformational leadership and employee creativity, important references such as Jyoty and Dev (2015), Tse, To, and Hiu (2018), Henker, Sonnentag, and Unger (2015), Nguyen, Nguyen, and Duong (2022), etc. are missing. The same is true for the other variables. The literature review is shallow and poor.

**Reviewer #2:**  he study makes a valuable contribution to the literature on organizational behavior and creativity, particularly within the context of Chinese SMEs in the manufacturing sector. The focus on creative self-efficacy (CSE) as a mediator adds theoretical depth and novelty to the research.

The choice of context (SMEs in China) is timely and appropriate, given the increasing global interest in innovation-driven growth in emerging markets.

**Reviewer #3:**  To ensure the scientific rigor and publication readiness of this study, the following revisions must be incorporated:

1. Further clarify the causal relationship path between "Individual Learning Orientation" (ILO) and "creative self-efficacy" (CSE).

2. Supplement the explanations of sample characteristics and data representativeness to enhance the external validity of the study. The article mentions that the data comes from 406 employees in small and medium-sized manufacturing enterprises in China, but it does not elaborate on the demographic characteristics of the sample (such as age, gender, job type, educational level, etc.), nor does it discuss whether the sample is representative.

**Do you want your identity to be public for this peer review?** For information about this choice, including consent withdrawal, please see our Privacy Policy

Reviewer #1: No

Reviewer #2: No

Reviewer #3: No

---

## [Author Response · Author response to Decision Letter 1]

8 Sep 2025

We would like to express our sincere gratitude for your time, effort, and insightful feedback on our manuscript. Your constructive comments were incredibly valuable and have helped us significantly improve the quality of our paper.

We have carefully and seriously considered every comment and have thoroughly revised the manuscript based on your excellent suggestions. A detailed, point-by-point reply to each comment can be found in the attached document titled "Response to Reviewers".

Thank you once again for your guidance and for the opportunity to strengthen our work. Best wishes!!

---

## [Decision Letter · Decision Letter 1]

2 Oct 2025

Unlocking employee creativity: How learning orientation and transformational leadership spark innovation through creative self-efficacy

PONE-D-25-10156R1

Dear Dr. Zeng,

We’re pleased to inform you that your manuscript has been judged scientifically suitable for publication and will be formally accepted for publication once it meets all outstanding technical requirements.

Kind regards,

Cheong Kim

Academic Editor

PLOS ONE

Additional Editor Comments (optional):

Reviewers' comments:

Reviewer's Responses to Questions

**Comments to the Author**

Reviewer #1: All comments have been addressed

2. Is the manuscript technically sound, and do the data support the conclusions?

Reviewer #1: Yes

3. Has the statistical analysis been performed appropriately and rigorously?

Reviewer #1: Yes

4. Have the authors made all data underlying the findings in their manuscript fully available?

Reviewer #1: Yes

5. Is the manuscript presented in an intelligible fashion and written in standard English?

Reviewer #1: Yes

Reviewer #1: In my opinion, the authors have improved the paper by following the reviewers' recommendations, and a significant change is noticeable. In my opinion, the article deserves to be published, although I would like to comment on the following details:

1. I would ask the authors to add a couple of sentences to the introduction to better respond to the comment I made in the first review: "The paper does not describe why it is important to study employee creativity. What does this study contribute to science and society?"

2. There is a typo in line 64: ‘creative outcomes. n this context’.

**Do you want your identity to be public for this peer review?** For information about this choice, including consent withdrawal, please see our Privacy Policy

Reviewer #1: No

---

## [Editor Report · Acceptance letter]

PONE-D-25-10156R1

PLOS ONE

Dear Dr. Zeng,

I'm pleased to inform you that your manuscript has been deemed suitable for publication in PLOS ONE. Congratulations! Your manuscript is now being handed over to our production team.

Kind regards,

on behalf of

Dr. Cheong Kim

Academic Editor

PLOS ONE